# Fabric Composites Reinforced with Thermally Bonded and Irregularly Aligned Filaments: Preparation and Puncture Resistant Performance

**DOI:** 10.3390/polym11040706

**Published:** 2019-04-17

**Authors:** Yu-Chun Chuang, Limin Bao, Mei-Chen Lin, Ting An Lin, Ching-Wen Lou

**Affiliations:** 1Interdisciplinary Graduate School of Science and Technology, Shinshu University, Nagano Prefecture 390-8621, Japan; yuchun780116@gmail.com (Y.-C.C.); baolimin@shinshu-u.ac.jp (L.B.); 2Department of Fiber and Composite Materials, Feng Chia University, Taichung City 40742, Taiwan; ritalin2870@gmail.com; 3Department of Science and Technology, Graduate School of Medicine, Science and Technology, Shinshu University, Nagano Prefecture 390-8621, Japan; 4Department of Medical Research, China Medical University Hospital, China Medical University, Taichung City 40402, Taiwan; 5Department of Bioinformatics and Medical Engineering, Asia University, Taichung City 41354, Taiwan; 6Innovation Platform of Intelligent and Energy-Saving Textiles, School of Textiles, Tianjin Polytechnic University, Tianjin 300387, China; 7College of Textile and Clothing, Qingdao University, Qingdao 266071, China; 8Department of Chemical Engineering and Materials, Ocean College, Minjiang University, Fujian 350108, China

**Keywords:** filament reinforcement, recycled aramid fabrics, thermal-bonding, puncture resistance

## Abstract

This study proposes fabric composites with improved static and dynamic puncture via increasing a friction force to restrain the slide of filaments as well as the compression and abrasion between the fibers and the puncture probe. The the bi-layered shell layers of composite fabrics are composed of aramid staple fibers and nylon staple fibers and a layer of low-melting-point polyester (LPET). The nonwoven layer consisting of recycled aramid and nylon staple fibers provides a shear effect to dissipate part of the puncture energy. Reinforcing interlayers include a woven fabric and PET filaments that are circularly aggregated between the surface layers, providing isotropic filament reinforcement and strengthening the resistance against the tip of the puncture probe. The reinforcing filaments may slide after the employment of needle punching, and to compensate for this disadvantage, the LPET layers are used to thermal bond the composite fabrics and the total thickness is controlled at 2 mm. The thermally bonded fabric composites are evaluated in terms of puncture resistance, thereby examining the effects of fabric structure and thermal bonding. According to the test results, the optimal composite structure is the sample N/L/W/F/L/N, which was reinforced by the LPET adhesive layer and irregularly aligned filaments. The sample which used the LPET adhesive layer had a positive influence on static puncture resistance and dynamic puncture resistance, preventing the slide of filaments, but the poor interfacial combination only contributed to limited reinforcement.

## 1. Introduction

Owing to the increasing awareness of the importance of self-protection, protective materials are being efficiently developed, including barks, hides, leather, metal, and composites. The progressing technology gives rise to diverse industries and subsequently demands for the preparation of protective materials that require large amounts of advanced materials have risen [1,2,3,4], and stab resistant materials are the most pervasively used in different fields [5,6]. Statistically, stabs, cuts, and abrasions account for 15% of workplace injuries per year, which makes the development of puncture resistant products utterly essential. In addition to being used for personal protection, stab resistant products can also be used in clothing textiles, household textiles, and industrial textiles. However, the need for these products to have both low cost and high performance restricts the common application of puncture resistant products (e.g., geotextiles and other industrial protective materials).

In light of the material types and purposes, puncture resistant materials can be divided into flexible, semi-flexible, rigid, and liquid stab resistant materials. Ceramic and metal plates are commonly used rigid stab resistant materials. They have excellent stab resistance but are heavyweight, which restricts the mobility of users. By contrast, flexible or semi-flexible stab resistant materials, such as high-density polyethylene, aramid mixture fabric, and specially made composites [7], have good stab resistance, are lightweight and flexible, but require multi-layered lamination to attain a good stab resistance. Hence, many researchers still anticipate creating puncture resistant materials that can be lightweight and still attain good performance. Moreover, shear thickening fluid (STF) is one of the most popular materials with stab resistance [8,9,10]. As evidenced by previous studies, a combination of fabrics and STF can obtain highly strengthened stab-proof and bulletproof functions [11,12,13,14]. However, there are some studies that proved that it was the STF particles which increased the friction force between fabrics and yarns, thereby improving the stab resistance of the STF-based fabrics [15,16,17].

To sum up, achieving good puncture resistance is mainly based on sufficient fiber density, high friction among fibers, a profound aggregation of fibers where the spike strikes, and firmly secured fibers. Hence, regardless of the type of stab resistant material, friction is one of the major factors influencing its stab resistance. This study aims to develop a flexible sandwich-structured puncture resistant fabric composites which can be used in the protective clothing field and geotextiles field. One of the materials we used in this study was the waste selvages of aramid plain woven fabrics (K129 and K29). Aramid fiber can usually be used in a wide variety of applications due to its light weight, high performance, and toughness. It is used in a variety of protective clothing and equipment to make it safer and more durable. Although aramid waste selvages can only be recycled into staple fibers, nonwoven fabrics can still take advantage of their original characteristics while reducing costs.

On the other hand, based on the fabric structure, knitted/woven fabrics have greater stab resistance against knives while nonwoven fabrics have puncture resistance against spikes [18,19,20,21]. Therefore, by combining the advantages of fibers and fabric structures, this study laminates filaments as a reinforcing interlayer to form the puncture resistant sandwich-structured fabric composites. A previous study shows that adding filament remarkably reinforces the puncture resistance of the composites, but the disadvantage is that the filaments slide severely. Hence, low-melting-point polyester (LPET) layers are used in order to improve this drawback of filament reinforcement. The employment of thermal treatment then secures the reinforcing filaments and mitigates the slide of the filament, thereby achieving greater stab resistance. According to the test results and compared with other studies, using the LPET adhesive layer has a positive influence, preventing the slide of filaments. At the same time, the internal porosity of the fabric is reduced, the utilization ratio of the fiber to the puncture resistance is improved, and the stability of the stab resistance of the composite fabric is improved.

Besides this, we used the nonwoven fabric process in this whole study. This process technology has the advantages of being fast, low cost, and having a high output, and can fully mix two or more kinds of fiber materials and composite multi-layer fabrics to make the flexible sandwich-structured puncture resistant fabric composites we designed.

## 2. Materials and Methods

### 2.1. Materials

Recycled aramid staple fibers (K129 and K29, provided by Jinsor-Tech Industrial Corporation, Taichung City, Taiwan) are obtained from aramid woven fabrics and have a fineness of 1.2 D and a length of 50–65 mm. High strength nylon 6 staple fibers (Formosa Chemicals & Fibre Corporation, Taipei City, Taiwan) have a fineness of 6.0 D, a length of 64 mm, single fiber strength of 10 g/d, and an elongation of 24.7%. Two-component low-melting-point polyester (LPET) staple fibers (Far Eastern New Century Corporation, Taipei City, Taiwan) have a fineness of 4.0 D and a length of 51 mm. The melting temperatures of the sheath and the core are 110 °C and 265 °C, respectively. Basalt woven fabrics (Yurak International, Taichung City, Taiwan) are composed of basalt fiber bundles at both warp and weft directions with a fineness of 2970 D and an areal density of 328 g/m^2^. PET continue filaments (Universal Textile, Taiwan) have a fineness of 500 D.

The commercially available geotextile is supplied by Hsinnjy Ltd. Co. (Taichung City, Taiwan), and the main material is PET fiber, which is a nonwoven fabric formed by multiple needle rolling and heat treatment. Its characteristics are based on the relevant test standards of this study. The basis weight is 272.46 g/m^2^; the thickness is about 1 mm, the tearing strength force is 299.53 N at the cross machine direction (CD) and 291.75 N at the machine direction (MD), the static puncture strength is 55.90 N, and the dynamic puncture strength is 35.52 N.

### 2.2. Preparation of Filament/Woven-Reinforced Fabric Composites

The symmetrically sandwich-structured composite fabrics are composed of double-layered surfaces and a reinforcing interlayer of a woven fabric and/or irregularly aligned filaments. As shown Figure 1, the surface layers are nylon/aramid recycled nonwoven fabrics (with an areal density of 200 g/m^2^) and pure LPET fabrics (with an areal density of 200 g/m^2^). Waste aramid selvages are processed by opening (fiber opening machine, TYM-40, HongChio Machinery Co., Ltd., Taoyuan, Taiwan) to have staple fibers, after which they and nylon staple fibers are needle punched to form the nonwoven layers (needle punching machine, SNP120SH6, Shoou Shyng Machinery Co., Ltd., New Taipei City, Taiwan). Pure LPET fabrics are composed of LPET fibers, serving as the adhesion layer to stabilize the composite structure. All of the laminates are needle punched (needle punching machine, SNP120SH6, Shoou Shyng Machinery Co., Ltd., New Taipei City, Taiwan) and thermally treated at 130 °C at a speed of 0.2 m/min (two-wheel hot press machine, CW-NEB, Chiefwell Engineering Co., Ltd., New Taipei City, Taiwan), forming the filamentor and woven-reinforced fabric composites. The first and second sub-figures in Figure 1 show the needle punching process and thermally treated process for forming the woven-reinforced fabric composites (N/L/W/L/N). The third sub-figure in Figure 1 shows the PET continuous filament lamination process which forms the filament-reinforced fabric composites (N/L/F/L/N). The total thickness of these sample are controlled at 2 mm. According to the materials we used in the reinforcing interlayer, the experimental groups are denoted as N/L/W/L/N (the first sub-figure in Figure 1), N/L/F/L/N (the third sub-figure in Figure 1), and N/L/W/F/L/N where “N” stands for nonwoven layer, “L” stands for LPET layer, “W” stands for woven interlayer, “F” stands for filament interlayer. The control group is thus denoted as N/L/L/N without reinforcing layers. The puncture resistance of the fabric composites is characterized in terms of their tear strength, static puncture resistance, and dynamic puncture resistance.

### 2.3. Tests

#### 2.3.1. Tearing Strength Test

As specified in ASTM D5035-11 (2015) [22], fabric composites were tested for tearing strength using an Instron 5566 (Instron, Canton, MA, USA). As shown Figure 2, the test was conducted using a constant tearing rate of 300 ± 10 mm/min. The distance between the upper and lower fixtures was 25.4 mm. Samples had a size of 150 mm × 75 mm and a 15 mm long notch. Ten samples for each specification were taken along the cross machine direction (CD) and machine direction (MD), respectively.

#### 2.3.2. Static Puncture Resistance Test

The static puncture resistance of samples was measured at a puncture rate of 508 mm/min using a universal strength testing machine (Instron5566, Canton, MA, USA) as specified in ASTM F1342 [23]. Samples had a size of 100 mm × 100 mm. The diameter of the puncture probe was 4.5 mm (as shown in Figure 3) and was chosen to be similar to an ice pick. Six samples for each specification were used for the test in order to have the average static puncture resistance, standard deviation, and coefficient of variation.

#### 2.3.3. Dynamic Puncture Resistance Test

The dynamic puncture resistance of samples was measured by a drop-weight impact machine (Kuang Neng Factory, Taiwan) that was equipped with a data collector (PCD300A, San Lien Technology, Taipei City, Taiwan) according to the energy level of E1-1 (24 J) in the NIJ Standard-0115.00 [24]. The puncture needle device contained a metal weight and a puncture probe, the total weight was 2.8 kg and the puncture probe was released from a height of 284 mm and fell freely (as shown in Figure 4). The maximum dynamic puncture resistance was obtained when the puncture probe penetrated the sample. Six samples for each specification were used for the test in order to have the average dynamic puncture resistance, standard deviation, and coefficient of variation.

## 3. Results

### 3.1. Tearing Strength

Figure 5 shows the tearing strength of the non-thermally treated filament/woven-reinforced fabric composites (denoted as NH) and Figure 6 shows the tearing strength of thermally treated ones (denoted as YH). The tearing strength along the CD of N/L/W/F/L/N was higher than that of the other experimental groups (i.e., N/L/W/L/N, N/L/W/L/N) and the control group (i.e., N/L/L/N). Moreover, N/L/W/F/L/N had a greater tearing strength after it was thermally treated. N/L/W/L/N is woven-reinforced and plain woven fabric has a regular plain arrangement, whereas N/L/F/L/N is filament-reinforced and the filaments are irregularly aligned. Unlike woven fabrics that have high fiber-yarn and yarn-yarn friction, filaments tended to slide during the tearing test. Additionally, N/L/W/F/L/N had many thermal bonding points as a result of the heat treatment, which in turn effectively improved the phenomenon of filament slide and thus achieved a higher tearing strength of 1144.0 N (Figure 6).

### 3.2. Static Puncture Strength

Figure 7 shows the static puncture resistance of filament/woven-reinforced fabric composites while Figure 8 shows their force-displacement curves. The non-thermally treated filament/woven-reinforced fabric composites are denoted as NH and the tearing strength of thermally treated ones is denoted as YH. Due to different features of the reinforcing woven- and filament-interlayers, N/L/W/F/L/N exhibited static puncture resistance in a combined manner. The shear force and friction between the probe and sample increased when the pointed probe further touched the sample (Figure 8). The resistance force against the probe sharply decreased (Figure 8a,b) right after the probe fully penetrated the sample (Figure 8c), leaving only the friction between the probe and the sample (Figure 8d). At the same time, the yarn-fabric structure of the samples underwent displacement, deformation, and final penetration damage, which makes the yarn-fiber friction and the slide of filament the two most important factors in this test. By comparison, the regularly formed plain-weave woven fabric provided greater puncture resistance than the irregularly aligned filaments for N/L/W/F/L/N. Furthermore, the thermal treatment generated thermal bonding points that can secure the whole structure of the fabric composites, preventing the slide of filaments. Unlike the woven fabrics with a fixed fabric pattern, filaments are freely aligned in loops as reinforcement and the slide of filaments adversely affects the static puncture resistance of thermally treated N/L/F/L/N. By contrast, N/L/W/F/L/N consists of both woven fabric and filaments as the reinforcing interlayers, both of which can be effectively combined. The irregularly arranged filaments can further improve the static puncture resistance at 0° and 90°. In particular, the thermally treated N/L/W/F/L/N had the maximum static puncture resistance of 243.2 N.

### 3.3. Dynamic Puncture Strength

Figure 9 shows the dynamic puncture resistance of N/L/W/F/L/N. Although both static and dynamic puncture strength tests used spike-shaped probes, the former used a constant rate of 508 mm/min, while the latter was added with an 8.5 kg weight and released from a specified height to plummet. Hence, the dynamic puncture resistance was demonstrated in a different way. Without thermal treatment, all of the samples had comparable dynamic puncture resistance. Like its static puncture resistance, the thermally treated N/L/W/F/L/N also outperformed the other groups and had the optimal dynamic puncture resistance. The employment of thermal treatment made a remarkable contribution due to the presence of thermal bonding points that significantly secured the sandwich structure and prevented the slide of filaments, strengthening the compound laminates firmly. Specifically, the thermally treated N/L/W/F/L/N had the maximum dynamic puncture resistance of 104.7 N.

### 3.4. Summary of Comparison

Table 1 shows the comparison of the proposed fabric composites, commercially available geotextiles, and the product of our previous studies [25]. Commercially available geotextiles have static and dynamic puncture resistances of 60 N and 37 N, indicating that the proposed N/L/W/F/L/N sample has greater puncture resistance. Similarly, the N/L/W/F/L/N sample also outperformed the proposed products of our previous study in terms of static and dynamic puncture resistance. Conversely, pure filament-reinforced (N/L/F/L/N) did not exhibit significant puncture resistance as the filaments inevitably slide, indicating that a bonding layer of multi-layered composites can restrain the fibers from sliding. In addition, PET filaments provided a much lower reinforcement than that of nonwoven fabrics and thermal bonding points. Despite the presence of an LPET adhesive layer, the combination of PET filaments and an LPET layer only provided a limited reinforcing effect. Hence, the future study will focus on the interfacial bonding force between the adhesive layer and the reinforcing layer. In terms of the EN 388:2016 standard, the puncture resistance of N/L/W/F/L/N reached level 3, which is between 100 N and 150 N and indicates good puncture resistance.

## 4. Conclusions

This study aims to develop sandwich-structured puncture resistant fabric composites, and the optimal N/L/W/F/L/N has good puncture resistance at level 3. The proposed fabric composites are characterized in terms of tearing strength, static puncture resistance, and dynamic puncture resistance. The test results indicate that thermally treated N/L/W/F/L/N has a 63.3% greater tearing strength at CD orientation, 63.9% greater static puncture resistance, and 32.5% greater bursting strength than N/L/L/N.

Compared to other studies, using the LPET adhesive layer had a positive influence, preventing the slide of the filaments, but the poor interfacial combination only contributes limited reinforcement. Hence, improving the interfacial affinity between laminates is suggested to be conducted in future studies.

## Figures and Tables

**Figure 1 polymers-11-00706-f001:**
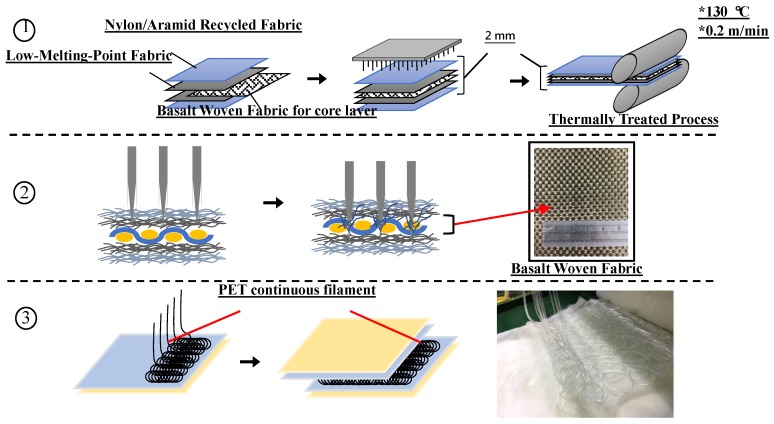
Schematic diagrams of the manufacture of fabric composites. (**1**) The laminates are needle punched and thermally treated to form the woven-reinforced fabric composites (N/L/W/L/N). (**2**) Sectional view during the needle punch process (N/L/W/L/N). (**3**) PET continuous filament lamination processing to form the N/L/F/L/N.

**Figure 2 polymers-11-00706-f002:**
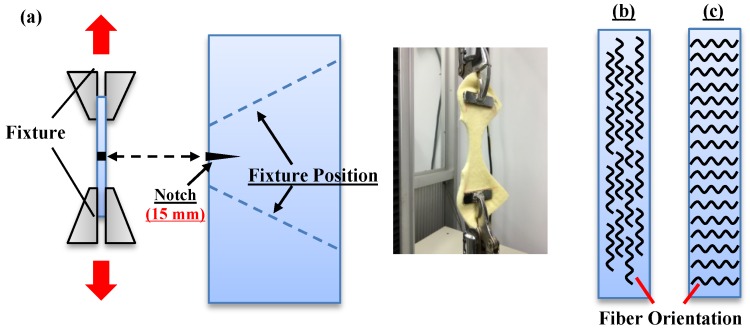
Schematic diagrams of the manufacture of fabric composites. (**a**) Test schematic of the tearing strength test; (**b**) Cross Machine Direction (CD); (**c**) Machine Direction (MD).

**Figure 3 polymers-11-00706-f003:**
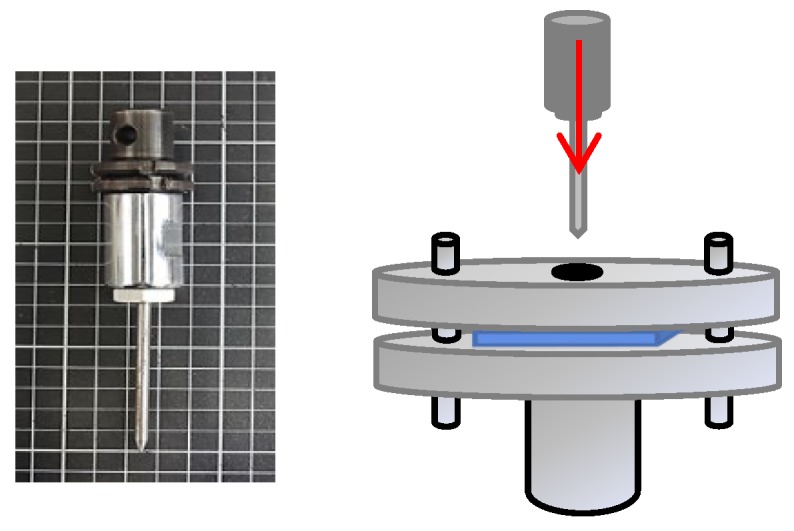
The equipment and puncture needle of static puncture resistance test.

**Figure 4 polymers-11-00706-f004:**
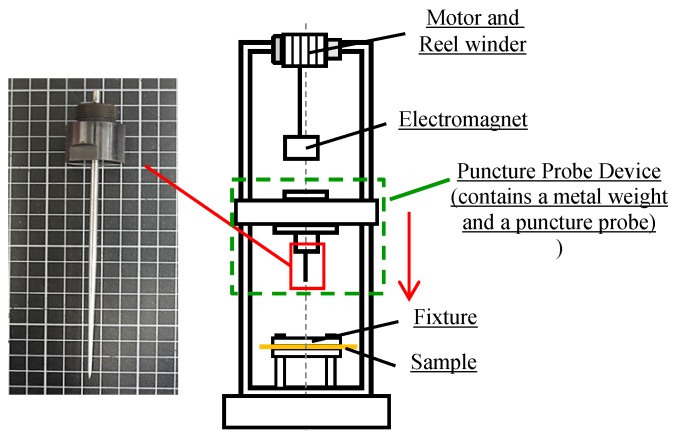
The equipment and puncture needle of dynamic puncture resistance test.

**Figure 5 polymers-11-00706-f005:**
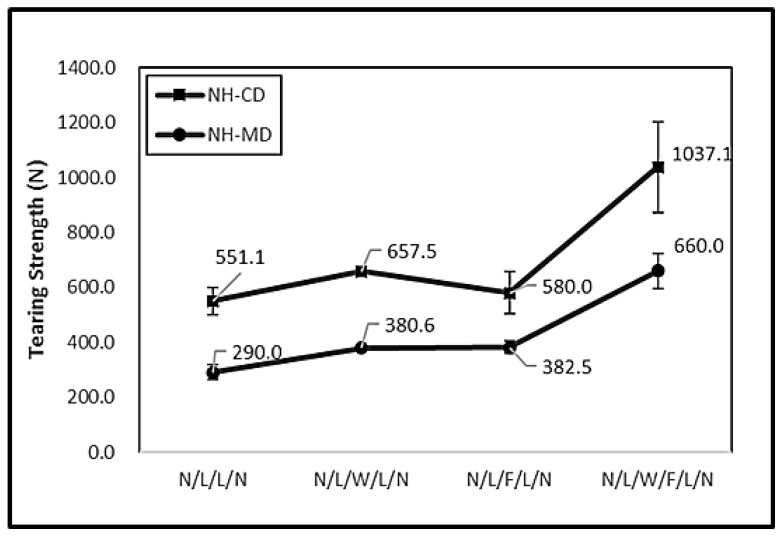
Tearing strength of filament/woven-reinforced fabric composites without heat treatment.

**Figure 6 polymers-11-00706-f006:**
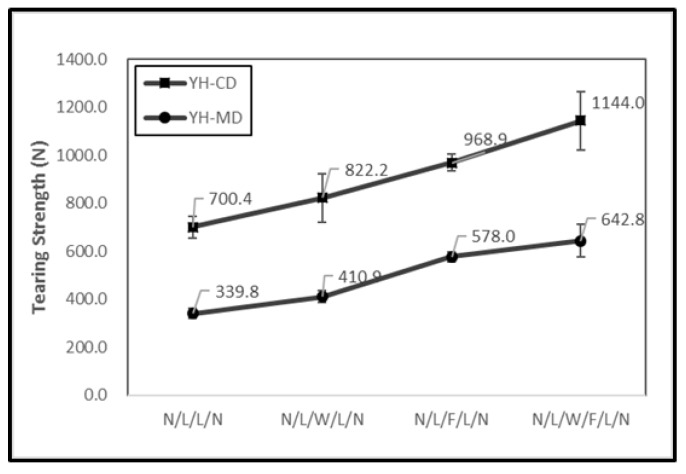
Tearing strength of filament/woven-reinforced composites fabrics that have been thermally treated.

**Figure 7 polymers-11-00706-f007:**
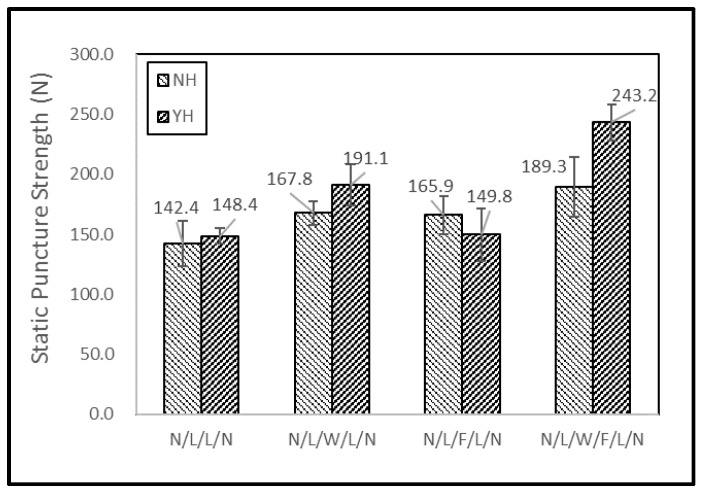
Static puncture resistance of filament/woven-reinforced fabric composites. (NH: non-thermally treated, YH: thermally treated).

**Figure 8 polymers-11-00706-f008:**
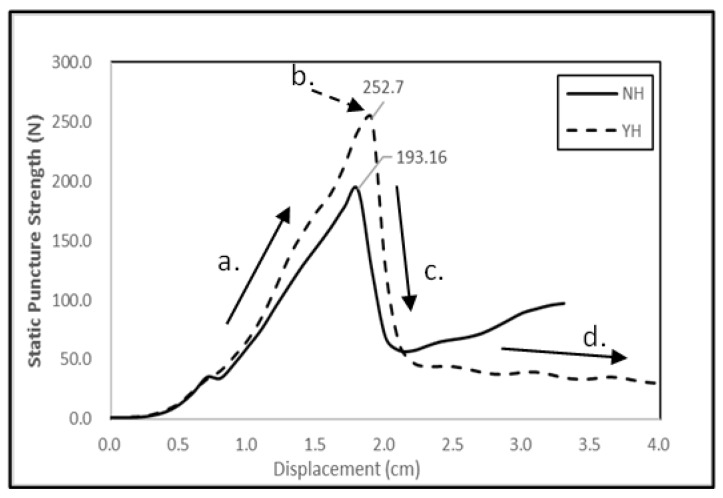
The strength-displacement curve of the static puncture resistance of filament/woven-reinforced fabric composites.

**Figure 9 polymers-11-00706-f009:**
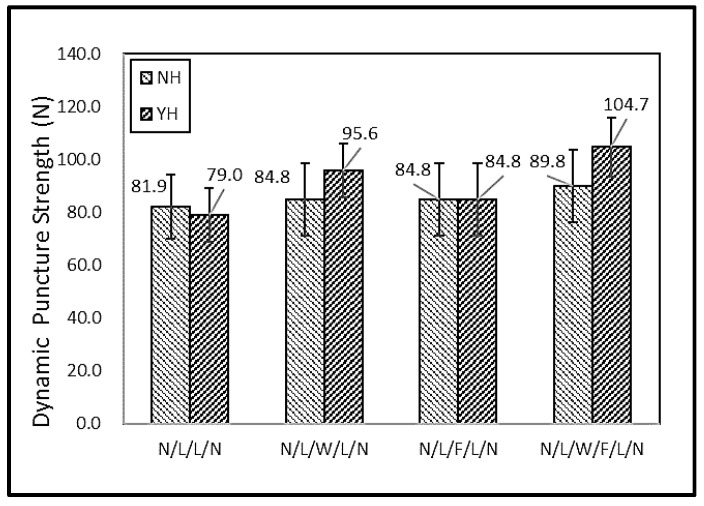
Dynamic puncture resistance of filament/woven-reinforced fabric composites. (NH: non-thermally treated, YH: thermally treated).

**Table 1 polymers-11-00706-t001:** Comparison of puncture resistances between N/L/W/F/L/N sample and other products.

	Static Puncture (N)		Dynamic Puncture (N)	
	Double-Layered Matrices	Matrix/F/Matrix	Double-Layered Matrices	Matrix/F/Matrix
Commercially Available Geotextiles	≈ 55.9		≈ 35.5	
Our Sample	148.4	149.4	79.0	84.8
EN 388:2016 Performance Levels	3 (100 < * < 150)	3 (100 < * < 150)	-	-

Note. “*” is the proposed the filament/woven-reinforced fabric composite.

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
