# Peer review of "Fabric Composites Reinforced with Thermally Bonded and Irregularly Aligned Filaments: Preparation and Puncture Resistant Performance"

_polymers, 2019, doi:10.3390/polym11040706_

Round 1

Reviewer 1 Report

Dear authors,

The manuscript requires some improvements and changes before I recommend it for publication in Polymers journal. The paper cannot be accepted for publishing in this form because some steps in manufacturing and conducting of the experiments are not very clear for the readers.

In order the authors improve the manuscript, I make the following remarks and comments:

1.Section “Abstract”:

-The main conclusion of the paper regarding the puncture strength level corresponding to the optimal composite structure N/L/W/F/L/N, is not written in abstract.

-  “The bi-layered surface layers” could be replaced with “the bi-layered shell layers” or “the bi-layered external layers”.

2. Section “1.Introduction”:

-What are the application fields (industry) for the composite fabrics involved in this study? What kind of products?

- The purpose and the main objectives of the paper should be much more clarified in the end of Introduction section.

3. Section “2.Materials and Methods” – Materials:

-Aramid fibers is a general name that includes Kevlar fibers. Please, correlate the text “Recycled Aramid staple fibers” with the name used in Fig. 1.

-What kind of Kevlar fibers is used? There are more kinds of Kevlar fibers depending on the chemical composition. Their mechanical characteristics are different. For material reproduction it is important to know the type of Kevlar fibers. What are their mechanical properties?

-Add information regarding the density on unit surface corresponding to the non-woven layers (Nylon/ Kevlar fabric, LPET - low-melting-point polyester) and basalt woven fabric.

-Please, add information about the basalt woven fabric like the following: density, type of yarns used on warp and weft directions (are different?) and so forth.

-Line 91: Some corrections have to be made in the text “… N/L/W/L/N, N/L/W/L/N, and N/L/W/F/L/N …”.

-Fig. 1: The text from this figure should be correlated with the related text. Please, add text in order to explain this figure. Write in caption of the figure what is denoted with 1 and 2. In the third sub-figure of Fig. 1 is not very clear if thermal treatment is or is not applied.

-Fig. 1: Write Basalt 79 woven fabric for core layer.

-Fig. 1: Explain that the first 3 sub-figures of 2 refers to the composite having woven fabric in the core while the next 3 sub-figures are related to the core consisting in filaments. Is not very clear what kinds of filaments are used. (PET filaments? Are continue or short filaments?)

Section “Tests”:

-It is a sub-section of the section “2.Materials and Methods”. So, delete “3” written in front.

- Please add in References section the following standards: ASTM D5035-11(2015), ASTM F1342, NIJ Standard-0115.00.

-Tearing Strength Test: I think that the authors would refer to the depth of the notch in the following text: “notch of a length of 15 mm” (line 103).

- Static Puncture Resistance Test: Is the diameter 4.5 mm of the puncture probe in accordance with the standard?

-Dynamic Puncture Resistance Test: Is puncture probe 118 weighing 2.8 kg recommended in standard in accordance with what property? (Thickness?)

What is the size specified height from which the puncture probe is released? Was the height the same for all fabrics involved in study?

Section “3. Results”:

-The abbreviation “NH” and “YH” used in Figures 4-8, should be explained in the text.

Section “4. Conclusion”:

The authors should add some conclusions regarding mechanical properties corresponding to the optimal structure N/L/W/F/L/N of the composite fabric. Compare the properties by expressing in percent the increasing of the properties compared to the ones corresponding to the other fabrics tested in this work.

Author Response

R1

Dear authors,

The manuscript requires some improvements and changes before I recommend it for publication in Polymers journal. The paper cannot be accepted for publishing in this form because some steps in manufacturing and conducting of the experiments are not very clear for the readers.

In order the authors improve the manuscript, I make the following remarks and comments:

1.Section “Abstract”:

-The main conclusion of the paper regarding the puncture strength level corresponding to the optimal composite structure N/L/W/F/L/N, is not written in abstract.

Ans: Thank you for the reviewer’s advice. We have revised the abstract  and added more narrative.

-The bi-layered surface layers” could be replaced with “the bi-layered shell layers” or “the bi-layered external layers”.

Ans: Thank you for the reviewer’s advice. We have revised the abstract section, added the experimental results and “the bi-layered surface layers” was replaced with “the bi-layered shell layers”.

2. Section “1.Introduction”:

-What are the application fields (industry) for the composite fabrics involved in this study? What kind of products?

- The purpose and the main objectives of the paper should be much more clarified in the end of Introduction section.

Ans: Thank you for the reviewer’s advice. We have revised the introduction section and added more research design details and research goals in the end of Introduction section.

3. Section “2.Materials and Methods” – Materials:

-Aramid fibers is a general name that includes Kevlar fibers. Please, correlate the text “Recycled Aramid staple fibers” with the name used in Fig. 1.

Ans: Thank you for the reviewer’s advice. We have revised the Fig. 1, that “Recycled Kevlar staple fibers” was replaced with “Recycled Aramid staple fibers”.

-What kind of Kevlar fibers is used? There are more kinds of Kevlar fibers depending on the chemical composition. Their mechanical characteristics are different. For material reproduction it is important to know the type of Kevlar fibers. What are their mechanical properties?

Ans: Thank you for the reviewer’s advice. The Kevlar material we used in this study is from the waste selvages of Kevlar plain woven fabrics (K129 and K29). And we have revised the text and added more details information in the introduction section and the Materials and Methods section.

-Add information regarding the density on unit surface corresponding to the non-woven layers (Nylon/ Kevlar fabric, LPET - low-melting-point polyester) and basalt woven fabric.

-Please, add information about the basalt woven fabric like the following: density, type of yarns used on warp and weft directions (are different?) and so forth.

Ans: Thank you for the reviewer’s advice. we have revised the text and added more details information in the Materials and Methods section.

The areal density of Nylon/Aramid recycles nonwoven fabrics and pure LPET fabrics are 200 g/m2. The Basalt woven fabrics is composed of basalt fiber bundles at both of warp and weft directions with a fineness of 2970 D and the areal density of 328 g/m2.

-Line 91: Some corrections have to be made in the text “… N/L/W/L/N, N/L/W/L/N, and N/L/W/F/L/N …”.

Ans: Thank you for the reviewer’s advice. We have revised the manuscript and added more narrative in Materials and Methods section.

-Fig. 1: The text from this figure should be correlated with the related text. Please, add text in order to explain this figure. Write in caption of the figure what is denoted with 1 and 2. In the third sub-figure of Fig. 1 is not very clear if thermal treatment is or is not applied.

Ans: Thank you for the reviewer’s advice. We have revised the manuscript and figure 1 and added more narrative in the Materials and Methods section. And both of the woven-reinforced fabric composites and the filament-reinforced fabric composites had a thermally treated process. The third sub-figure of Fig. 1 is for showing the PET continuous filament lamination processing.

-Fig. 1: Write Basalt 79 woven fabric for core layer.

Ans: Thank you for the reviewer’s advice. We have revised the Fig. 1 and added more details and related with the text.

-Fig. 1: Explain that the first 3 sub-figures of 2 refers to the composite having woven fabric in the core while the next 3 sub-figures are related to the core consisting in filaments. Is not very clear what kinds of filaments are used. (PET filaments? Are continue or short filaments?)

Ans: Thank you for the reviewer’s advice. We have revised the Fig. 1 and added more details and related with the text.

Figure 1. Schematic diagrams of the manufacture of fabric composites. (1) the laminates are the needle-punched process and thermally treated process for forming the woven-reinforced fabric composites(N/L/W/L/N). (2) sectional view during needle-punched processing(N/L/W/L/N). (3) PET continuous filament lamination processing for forming the N/L/F/L/N.

Section “Tests”:

-It is a sub-section of the section “2.Materials and Methods”. So, delete “3” written in front.

Ans: Thank you for the reviewer’s advice. We have revised the text.

- Please add in References section the following standards: ASTM D5035-11(2015), ASTM F1342, NIJ Standard-0115.00.

Ans: Thank you for the reviewer’s advice. We have revised the references section.

-Tearing Strength Test: I think that the authors would refer to the depth of the notch in the following text: “notch of a length of 15 mm” (line 103).

Ans: Thank you for the reviewer’s advice. In the full text, we have revised the tearing test section and supplements the test schematic with test details and hopes to help the reviewer can understand more easily.

Figure 2. Schematic diagrams of the manufacture of fabric composites (a) Test schematic of the tearing strength test (b) Cross Machine Direction (CD) (c) Machine Direction (MD).

- Static Puncture Resistance Test: Is the diameter 4.5 mm of the puncture probe in accordance with the standard?

Ans: Thank you for the reviewer’s question. The test speed and device for the static puncture test tested in this study was set according to the standard ASTM F1342. The diameter of the puncture probe is chosen to be similar to the ice pick.

-Dynamic Puncture Resistance Test: Is puncture probe 118 weighing 2.8 kg recommended in standard in accordance with what property? (Thickness?)

Ans: Thank you for the reviewer’s question. According to the standard, the puncture needle device should contain a metal weight and a puncture probe, the total weight is 2.8 kg. And the puncture probe is released from a height of 284 mm and falls freely (as shown in figure 4). We have revised the text and added more details.

Figure 4. The equipment and puncture needle of dynamic puncture resistance test.

What is the size specified height from which the puncture probe is released? Was the height the same for all fabrics involved in study?

Ans: Thank you for the reviewer’s question. According to the standard, the puncture probe is released from a height of 284 mm and falls freely (as shown in figure 4). The total thickness of this fabric composite we designed is controlled at 2 mm. We have revised the text and added more details.

Section “3. Results”:

-The abbreviation “NH” and “YH” used in Figures 4-8, should be explained in the text.

Ans: Thank you for the reviewer’s advice. In the full text, we have revised and added narrative. The non-thermally treated filament/woven-reinforced fabric composites denoted as NH and the tearing strength of thermally treated ones denoted as YH.

Section “4. Conclusion”:

The authors should add some conclusions regarding mechanical properties corresponding to the optimal structure N/L/W/F/L/N of the composite fabric. Compare the properties by expressing in percent the increasing of the properties compared to the ones corresponding to the other fabrics tested in this work.

Ans: Thank you for the reviewer’s advice. We have revised the conclusion section and compare the properties by expressing the increase in the properties as a percentage.

Reviewer 2 Report

The authors of the paper performed the experimental study of fabrication and testing of fabric composite reinforced with adhesive layer for improving the puncture resistant performance. The research of the work is within the scope of the Journal with some new sample and test designs and original test data. The paper can be considered for publication in this Journal provided that the authors can further address the following issues.

Title

(a) The entire name of abbreviation "LPET' should be given.

(b) The title can be shorted in a more concise, physically meaningful way

2. Abstract

(a) The authors may provide some numerous data in the Abstract such as the composite thickness, test data, or any improvement compared to existing data in the literature, somehow.

3. Introduction

(a) The authors may need to cite some experimental data and existing puncture resistant laminate design available in the literature.

(b) The authors are suggested to highlight their composite design and give a few rational justifications such as cost, weight, puncture resistance, among others. It seems the authors just emphasized the literature work such as shear-thickening-fluid based materials, etc. 

2. Material methods

(a) it is not clear how much the thickness of each sub-layer of the laminate is?

(b) It will be better if the authors can provide a few clear images to show the sample

(c) The authors are suggested to summarize the fabrication procedure including the thermal treatment process, such as temperature, pressure, duration and hot-press machine used in the tests.

3. Test & Results 

(a) Give a diagram to show the Tearing Strength Test? Illustrate or clearly describe what the cross machine (CD) and machined direction (MD) are?

(b) Add a few images to show the samples before and after tests

(c) Besides the data shown in Figs. 4, 5, 6 and 8, it can be much better for the authors to add a few raw force-displacement diagrams obtained directly from tests, similar to Fig. 7, to show the first-hand experimental data before reduction. 

(d) It is hard to understand the legends "NH-CD, NH-MD, YH-CD, YH-MD, NH, YH" as they are not clearly defined in context or in the figures. The authors are suggested to articulate them for easier understanding.

(e) When the authors compared their puncture resistance force with those in the literature, they did not mention the basis for comparison such as testing methods, sample dimensions, materials, even costs for materials and manufacturing, etc. Without these, the comparison clearly does not have any value. 

4. Language 

(a) Largely, the writing is pretty good while a few spots exist for further improvement such as the last sentence of the 2nd paragraph in Page 2, among others.

Author Response

Please see the attached file, thank you!

Round 2

Reviewer 1 Report

Dear authors,

I reviewed again the revised version of your manuscript (version v2).

I could see that you made all changes in accordance with my recommendation and you also responded to my questions in the revised version.

In order to recommend your paper for publishing, please do the following little corrections:

-You used firstly the abbreviations CD and MD for cross machine direction and machine direction respectively, in line 134 of the revised version v2 of your manuscript. But you explain these abbreviations only in line 180. You must explain the abbreviation in Line 134 firstly.

-Fig. 2. Please add a dimension to indicate the length of the notch having 15 mm. Otherwise, the text related to Fig. 2 remains unclear to readers.

Author Response

R1-Comments and Suggestions for Authors

Dear authors,

I reviewed again the revised version of your manuscript (version v2).

I could see that you made all changes in accordance with my recommendation and you also responded to my questions in the revised version.

In order to recommend your paper for publishing, please do the following little corrections:

-You used firstly the abbreviations CD and MD for cross machine direction and machine direction respectively, in line 134 of the revised version v2 of your manuscript. But you explain these abbreviations only in line 180. You must explain the abbreviation in Line 134 firstly.

-Fig. 2. Please add a dimension to indicate the length of the notch having 15 mm. Otherwise, the text related to Fig. 2 remains unclear to readers.

Ans: Thank you for the reviewer’s advice. We have revised the manuscript and figure 2.

“…the tearing strength force is 299.53 N at the cross machine direction (CD) and 291.75 N at the machine direction (MD)…”

Reviewer 2 Report

The authors have revised the paper according largely to the review comments while the authors were unable to release the typical raw data (diagrams) of their tests under the request of the review comments. Thus, the reduced data, bar diagrams, and other results are questionable. This reviewer basically does not recommend the further consideration of the paper for publication in this journal while the final decision should be made by the editors.

Author Response

R2-Comments and Suggestions for Authors

The authors have revised the paper according largely to the review comments while the authors were unable to release the typical raw data (diagrams) of their tests under the request of the review comments. Thus, the reduced data, bar diagrams, and other results are questionable. This reviewer basically does not recommend the further consideration of the paper for publication in this journal while the final decision should be made by the editors.

Ans: Thank you for the reviewer’s much advice. And I am sorry for the reviewer’s comments. Obviously, the previous reply cannot convince the reviewer. Therefore, the raw data (diagrams) of the tearing test and dynamic puncture test are showing as below. Thanks again for the reviewer’s advice.

The force-displacement curve of the tearing test.

The load-time curve of dynamic puncture test. (Data capture frequency: 2000 Hz)
